# Mechanical Spectroscopy Study of CrNiCoFeMn High-Entropy Alloys

**DOI:** 10.3390/ma16103689

**Published:** 2023-05-12

**Authors:** Enrico Gianfranco Campari, Angelo Casagrande

**Affiliations:** 1Department of Physics and Astronomy, Alma Mater Studiorum-University of Bologna, Viale Berti Pichat 6/2, 40127 Bologna, Italy; 2Department of Industrial Engineering, Alma Mater Studiorum-University of Bologna, Viale Risorgimento 4, 40136 Bologna, Italy

**Keywords:** mechanical spectroscopy, dynamic Young’s modulus, selective laser melting, debye peak

## Abstract

The equiatomic high-entropy alloy of composition of CrNiCoFeMn with an FCC crystal structure was prepared by either induction melting or additive manufacturing with a selective laser melting (SLM) process, starting from mechanically alloyed powders. The as-produced samples of both kinds were cold worked, and in some cases re-crystallized. Unlike induction melting, there is a second phase, which is made of fine nitride and Cr-rich σ phase precipitates, in the as-produced SLM alloy. Young’s modulus and damping measurements, as a function of temperature in the 300–800 K range, were performed on the specimens that were cold-worked and/or re-crystallized. Young’s modulus values of (140 ± 10) GPa and (90 ± 10) GPa were measured from the resonance frequency of free-clamped bar-shaped samples at 300 K for the induction-melted and SLM samples, respectively. The room temperature values increased to (160 ± 10) GPa and (170 ± 10) GPa for the re-crystallized samples. The damping measurements showed two peaks, which were attributed to dislocation bending and grain-boundary sliding. The peaks were superposed on an increasing temperature background.

## 1. Introduction

Two papers were published in 2004 by B. Cantor [1] and J. W. Jeh [2] suggesting a new and completely different set of strategies for developing new materials. Despite the fact that the articles did not attract much attention immediately after their publication, they became, in less than a decade, very influential papers. The alloys they reported on were named high-entropy alloys (HEA) and Cantor’s alloys, after the name of the discoverer. These pioneering studies reported on an alloy that was made of the following five transition metals: Cr, Ni, Co, Fe, and Mn, which were crystallized as a single solid solution [1,2,3,4].

These HEAs had a concentration of each of the main elements in the range of 5–35 atomic % [2]. The alloys were characterized by a high mixing entropy in their liquid state [3] and could crystallize as a single phase (e.g., FCC, HCP, or BCC) instead of multiple intermetallic phases. Furthermore, they seemed to offer some promising technological features, such as increased hardness, good wear resistance, excellent strength at both high and low temperatures, and good properties of resistance to corrosion and oxidation [5,6,7,8]. The unique properties of the HEAs were ascribed to the inherent properties of multicomponent solid solution formation, such as distorted lattice structures, the cocktail effect, sluggish diffusion, and the formation of nanoscale deformation twins [9].

There is no doubt that the suppression of the brittle intermetallic phases in such alloys is a really interesting feature for metallurgical applications. As reported by other authors, the CrNiCoFeMn HEAs are considered to be stable disordered supersaturated FCC solid solutions with high ductility and remarkable fracture toughness [8]. They also usually exhibit good thermal stability [6,7,10]. Furthermore, processes related to diffusion, including grain growth and re-crystallization, are anticipated to be slow in HEAs.

The belief that the equiatomic CrNiCoFeMn alloy is an HEA example with a single disordered solid solution structure was recently challenged by the discovery of (precipitates) second phases in the alloy after low-temperature annealing. Nitrides and σ phase precipitates inside of the grains and at the grain boundaries may be present. Together with twins and other defects, they influence the mechanical properties of these alloys. Due to this, it is important to control the new phase precipitation or dissolution upon heat treatment. In addition, it was reported that HEAs with two or more phases can bring the strength–ductility trade-off under control [11,12].

Several studies in recent years have been aimed at determining the mechanical properties of these alloys [13,14,15,16,17,18]. In particular, yield strength, hardness, fracture toughness, strain hardening, and elastic constants were measured for alloys of various compositions and subjected to a number of thermal and mechanical treatments. As far as the elastic constants are concerned, the experimental measurements were mainly aimed at measuring Young’s modulus and the shear modulus, mostly using resonant ultrasound spectroscopy techniques, stress–strain tests, and nanoindentation. However, very few data are related to mechanical spectroscopy (MS) measurements. The term refers to an experimental technique based on joint measurements of damping and the dynamic elastic modulus [19,20]. Damping, also named internal friction, is defined as the capacity of a material to convert vibrational mechanical energy into heat. A sample is put into resonance vibration by means of some kind of excitation, such as electrostatic or mechanical. When the excitation is turned off, the oscillations with a decreasing amplitude are measured. The rate of amplitude decrease, as will be better explained in the following section, determines the damping value. Alternatively, forced oscillations are induced, and the lag between the stress and strain is measured. Internal processes that give rise to damping can often yield internal friction peaks as a function of temperature or frequency. Conversely, peak detection in an MS test is indicative of some dissipation phenomena in the material.

The Young’s modulus or shear modulus measured during an MS test, depending on which kind of stress is applied to the sample, are usually more precise and reliable than those that are obtained by static or indentation measurements. Besides measuring the modulus value, in an MS test, it is also possible to obtain its temperature dependence. Whenever an internal friction peak is observed, the moduli display a change (for details see [19], Chapter 3). In a temperature-dependent measurement, the modulus relaxation at the peak is superimposed to the usual decrease in temperature due to thermal expansion.

Anelasticity is the phenomenon at the basis of this technique. It is a stress–strain relationship of a body that is linear, unique (elastic behavior), and time-dependent. It is, therefore, different from viscoelasticity, which includes anelasticity as the special case of complete strain recoverability. The anelastic behavior may be due to several microscopic processes, which are both time-dependent and able to dissipate the mechanical energy that is stored in a sample. The MS technique is, therefore, extremely sensitive to the various features of the material microstructure, such as dislocations, grain boundaries, precipitates, twins, and internal stress. It is also non-destructive and supplies information that is not obtainable with other experimental setups. In several cases, the internal processes that give rise to damping are thermally activated and are characterized by a well-defined activation energy. Consequently, it is possible to study the damping capacity of a sample as a function of temperature, hence the term mechanical spectroscopy. A peak corresponds to each process, which is usually, but not always, a Debye peak.

In the case of Cantor’s alloys, peculiar anelastic effects could be exhibited due to the presence of twins or to the high degree of disorder. Furthermore, it is certainly interesting to compare alloys with the same composition that are produced in different ways. The aim of this study is thus to measure the MS behavior of Cantor’s alloys of the CrNiCoFeMn composition that have been prepared in two different ways, as described below. Specifically, the temperature-dependent damping and Young’s elastic modulus of two alloys that were produced by induction melting and laser melting, deformed by cold rolling, and re-crystallized were measured and correlated with their different microstructures. The test equipment is of the resonant bar type, that is, the specimen is mounted in single cantilever mode, and it is excited to its first longitudinal vibration mode by applying an alternating voltage. Resonances in the acoustic range were obtained.

## 2. Materials and Methods

### 2.1. Standard Production Process

As reported in greater detail in previous articles [21,22], and shown in Figure 1, a nominally equiatomic HEA was produced by two different processes. In the first and more standard process, Cr, Ni, Co, Fe, and Mn powders, supplied by Sigma Aldrich (Darmstadt, Germany), with a purity greater than 97% atomic, were subjected to pre-alloying by mechanical milling in an Argon atmosphere using a Planetary Ball Mill (PM 100 by Retsch GmbH, steel balls with BPR 15:1, Haan, Germany) working at 400 rpm. Treatment cycles lasting 15 min each, with a 5 min break between them, for a total grinding time of 45 h were used. The break time was used to avoid overheating. A vacuum induction furnace equipped with alumina crucibles was subsequently used to completely melt the alloyed powders. The resulting disks had a diameter of 35 mm, an 8 mm height, and weighed between 56 g and 58 g. Heating temperatures in excess of 1720 K and times of about 30 min were employed. Induction melting of mechanically pre-alloyed powders was used as a synthesis approach in order to reduce element segregation and Mn loss. The as-cast disks were cut into bars, which were cold rolled in the laboratory in order to reduce their thickness from 3 mm down to 0.3 mm. Some of the cold-rolled specimens were subsequently re-crystallized via 30 min annealing at 1173 K in a Kanthal Super HT rapid high-temperature furnace (Hallstahammar, Sweden). The sample’s density was ρ = (7.1 ± 0.2) g/cm^3^, without any significant difference between the as-cast and annealed samples.

### 2.2. Selective Laser Melting Production Process

In the second process, the same starting mechanically pre-alloyed powders were melted by selective laser melting (SLM) in additive manufacturing [23,24]. The SLM apparatus that was used to produce the samples with a volume of 50 × 5 × 5 mm^3^ was a SISMA MYSINT100 RM (Vicenza, Italy). A high purity nitrogen atmosphere was used in order to minimize oxidation during the production process. Melting did not take place until the oxygen level dropped below the set limit threshold, which was 0.5%. The sample’s surface perpendicular to the growth direction was divided into 6 slices, filled according to a chessboard strategy [25]. Again, sections of the starting material were cold rolled with a 90% thickness reduction and re-crystallized via 30 min annealing at 1173 K. The sample densities were ρ = (7.0 ± 0.1) g/cm^3^ and ρ = (7.1 ± 0.1) g/cm^3^, before and after re-crystallization, respectively.

### 2.3. Testing Techniques

All specimens were characterized as follows:(a)Micro-structural investigations. They were performed by either electron microscopy or optical microscopy. For the latter, an Olympus GX71 (Tokyo, Japan) was used, while a Tescan MIRA3 (Brno, Czech Republic), provided with an energy dispersive microanalyzer (EDS) by Bruker Quantax (Billerica, MA, USA), was used for electron microscopy observations. The specimens used for these analyses were polished and chemically etched with a glyceregia solution (1 HNO_3_ + 3 HCl + 3 Glycerol). Electron backscattered diffraction (EBSD) was used to determine the grain size and crystallographic orientation. EBSD maps were obtained with a Quantax EBSD detector to document the FCC matrix alloy and the precipitation of secondary phases. The EBSD data were recorded and analyzed using Bruker Esprit 2.1 software.(b)X-ray diffraction (XRD). The crystallographic structure was determined by the use of a Panalytical X’Pert PRO diffractometer equipped with a gas proportional detector (Malvern, UK). A Q/2Q scan was performed in the 2Q range from 0.6 to 1.75 rad. In order to eliminate sample displacement errors, and to achieve a correct unit of cell determination, a parallel beam configuration was used, including an X-ray mirror (incident beam optics) coupled with a long soller slit and a flat monochromator (diffracted beam optics).(c)Mechanical spectroscopy. Damping and dynamic modulus measurements were performed in a vacuum by means of the mechanical analyzer VRA 1604 [26,27]. In the VRA apparatus, specimens are mounted in the free-clamped mode and excited by flexural vibrations. The specimens were kept in resonance while the temperature changed at the selected rate. The resonance frequency of all specimens was in the 300 to 1000 Hz range and the strain amplitude was about 10^−5^. The specimens were heated from room temperature up to a maximum temperature of 800 K at a rate of 1.5 K/min. Electrostatic excitation was used to put the samples into resonance. The logarithmic decay of the flexural vibrations after turning off excitation was used to determine the damping parameter (usually referred to as *Q*^−1^) as follows:
(1)$$Q^{-1}=\frac{1}{m\pi }ln(\frac{A_{n}}{A_{n+m}})$$
where *A_n_* and *A_n_*_+*m*_ are the amplitudes of the *n*-th and (*n + m*)-th oscillation. The resonance frequency f allowed us to compute the dynamic modulus E at the same time, as follows:(2)$$E=\frac{48\pi^{2}PL}{m^{4}h^{2}}f^{2}$$
where *Ρ* is the material density, *L* is the length of the sample, *h* is the thickness, and *m* is a constant (*m* = 1.875). When the following condition is fulfilled, a Debye relaxation peak occurs:(3)$$\omega \tau =\omega \tau_{0}exp(\frac{H}{kT})=1$$
where *τ* is the relaxation time, *τ*_0_ is the pre-exponential factor, *ω* = 2*πf* is the angular frequency, *H* is the activation energy of the process giving rise to the peak, *T* is the temperature, and *k* is the Boltzmann constant.

## 3. Results

As reported in the introduction, the mechanical spectroscopy measurements obtained for Cantor’s alloys of the CrNiCoFeMn composition that were prepared in two different ways (in a deformed state by cold rolling and in a re-crystallized state) were compared. Cold working was used to increase the strength and promote re-crystallization, as is commonly performed for alloyed austenitic stainless steels [28,29], which chemically and mechanically resemble Cantor’s alloys. The improvement in mechanical properties such as hardness, fatigue strength, and tensile strength so obtained were predominantly due to the induction of compressive residual stress in the specimens [22].

The typical damping and dynamic modulus curves of CrNiCoFeMn produced by standard processes are reported in Figure 1. The curves of Figure 1 refer to laminated but not-re-crystallized samples. The experimental values that are reported in dark-red and red refer to a first thermal run, while those in black and gray refer to a second run on the same sample. In the first run, in regard to the damping, the presence of a dissipation peak was observed at T = 400 K, together with a growing background, with a Q^−1^ value of about 7.5 × 10^−4^ at T = 300 K, which grew up to 18 × 10^−4^ at 700 K. Similar values are commonly measured in many steels and superalloys. This peak was observed in all of the laminated samples, both standard and SLM. It disappeared completely after heating above 700 K. The dynamic modulus curve (normalized to the initial value, M_0_, at room temperature) showed a slope change to the Q^−1^ peak, superimposed on a monotonous decreasing trend. This is the modulus relaxation accompanying a peak due to an anelastic effect.

The 2nd run revealed how heating during the 1st run, here up to 700 K, and up to 800 K in other trials, yielded a damping decrease. This is a commonly observed relaxation effect due to the internal stress release caused by the heat treatment and shows that cold working introduces dislocations and internal stresses into the material, whose effect is to increase the damping and decrease the elastic modulus. The damping at T = 300 K decreased to 4 × 10^−4^. The disappearance of the peak at T = 400 K is quite interesting, which is evidently linked to microstructural features of the material that can be modified by heat treatments at relatively low temperatures. The same effect was also evident in the elastic modulus (normalized to the initial value, as in the 1st run), in which no modulus deficit appeared. The behavior displayed in the 2nd run was stable, in the sense that further heating up to 800 K (in some cases up to five runs were performed) did not induce changes in the Q^−1^ or the elastic modulus trends.

Figure 2 shows the damping data of Figure 1, analyzed in terms of an exponential increasing background and a Debye peak. In order to determine the peak temperature and the apparent activation energy, experimental *Q*^−1^ data have been fitted by considering a Debye peak, described by the following expression:(4)$$Q^{-1}=\Delta \cdot sech\left[\left(\frac{H}{k})\cdot (\frac{1}{T}-\frac{1}{T_{p}}\right)\right]$$
where *k* is the Boltzmann constant, Δ is the relaxation strength, *H* is the activation energy, and *T_p_* is the peak position.

A background was added to the Debye peak, as described by the following:(5)$$Q^{-1}=A+B\cdot exp(\frac{T}{C})$$
where *A* represents the temperature-independent damping term. In Table 1, the values of the fit parameters that were used for the Debye peak, plus the background, are reported. The relaxation strength is $\Delta =2\times 10^{-4}$, with activation energy $H=\left(0.5\pm 0.1\right)\;eV$ (~48 kJ/mol). An evaluation of the activation energy of the peak by means of the displacement of its maximum as a function of the frequency [19] (Chapter 3) was not accomplished due to the lack of data from samples having a sufficiently wide range of resonance frequencies.

A continuous damping contribution, often increasing exponentially with temperature, was measured in several metals. This background, on which peaks of various origin may be superposed, was higher in the cold-worked samples, while it decreased with grain size increase and after annealing at high temperatures. Despite some still controversial points, it is commonly accepted that the increasing background is the result of the contribution of a broad spectrum of diffusion-controlled relaxation processes [30,31]. The objects that give rise to the relaxation, grain boundaries, and dislocations interacting with point defects are considered to have the most significant contributions.

Typical damping and dynamic modulus curves of a standard CrNiCoFeMn specimen after re-crystallization are reported in Figure 3. The behavior is qualitatively the same as that of the not-re-crystallized samples after a first thermal run above 700 K. Q^−1^ monotonously increased with temperature, and no peaks are visible. Both damping and the modulus are stable and display the same behavior in successive runs up to 800 K.

The elastic modulus decrease, as a function of temperature (about 17% in the 300 to 800 K range), was comparable to the case of the 2nd run of the not-re-crystallized sample. However, its absolute value increased with respect to the case of the not-re-crystallized samples, with E = (140 ± 10) GPa in the case of the not-re-crystallized samples and E = (160 ± 10) GPa in the case of re-crystallized samples (room temperature values). The anisotropy of the elastic modulus and the texture change during re-crystallization [22] may be responsible for this increase [32], together with dislocation annihilation. In addition, the re-crystallization treatment modified the grain boundary structure, contributing, in turn, to the observed variation of damping and modulus.

Let us now take into consideration the samples produced by SLM. The temperature behavior for T < 600 K in a 1st run was the same as that reported in Figure 1 for the standard alloy. In both kinds of sample, a peak that disappeared after the first thermal heating was detected, leading to a reduced background and a stable behavior in the subsequent runs. Significant differences appeared in the 2nd run, as shown in Figure 4, where data relating to both a not-re-crystallized (black and gray) and a re-crystallized (dark-red and red) sample are reported. The comparison between the samples that were already heated up to 800 K is significant because it allows us to compare stable structures, at least in the temperature range that was used for the test. In the not-re-crystallized sample, a damping peak appeared at 685 K, superimposed on the usual growing background. The peak was stable when heating up to 800 K, but disappeared after re-crystallization, as shown (dark-red data) in Figure 4. The re-crystallization treatment also yielded a more relaxed structure than that of the samples that were treated only up to 800 K. A considerable difference was also measured in the elastic modulus, which increased from the relatively low value of E = (90 ± 10) GPa in the not-re-crystallized sample to E = (170 ± 10) GPa in the re-crystallized sample. The relatively low value measured in the SLM samples before re-crystallization was probably due to a microscopic and widespread porosity brought about by melting followed by rapid cooling, as already observed experimentally and modeled in other materials [33,34]. Diffusion and grain boundary rearrangement during re-crystallization modified the elastic modulus, raising its value towards the standard values. Figure 5 shows the damping curves of the re-crystallized sample of the previous figure, analyzed as before in terms of an exponential increasing background and Debye peak. In Table 1, the fitting parameter values are reported. The activation energy of this new peak, determined as before, turns out to be $H=\left(1.3\pm 0.1\right)\;eV$ (~127 kJ/mol).

In Figure 6, the diffraction pattern on an SLM sample after re-crystallization is reported. Peaks corresponding to the FCC phase of the Cantor’s alloy were detected, together with those of the precipitated phases. Precipitates were already present in the as-made material. The small size of these precipitates, however, did not provide a sufficient scattering volume to allow this phase to be detected by XRD diffraction. In the re-crystallized material, precipitates increased in size, decreased in number, and became visible. Figure 7 shows optical images of the as-produced samples, together with the cross-sectional images of a standard sample, before and after re-crystallization. The as-produced samples have a dendritic structure, and the re-crystallized samples have a diffuse twin structure. No precipitates were observed in the standard samples. Instead, they were detected in the samples produced by SLM, as can be seen in Figure 8, for the case of a cold-worked sample. Precipitates were already present before cold working [22]. Note how after re-crystallization, both in the standard and SLM samples, there was a widespread presence of geminates, together with a transition from elongated to more rounded grains (Figure 7b and Figure 8d).

## 4. Discussion

### 4.1. Low Temperature Peak of the CrNiCoFeMn Alloy after Cold Rolling, without Re-Crystallization

Even though it is important to examine how process parameters affect the microstructure in order to explain the damping capacity and the observed features of these alloys, there are limited experimental data available in this regard. In this work, the same starting mechanically pre-alloyed powders were used for both the standard fusion and the SLM process. For the latter, it was expected that the mechanical behavior would be affected by the stress field due to the SLM process. At the same time, the local heating under stress conditions was expected to cause strain relaxation in the surrounding areas. Further stress relaxation was mainly caused by an oriented slip of dislocations, by cold rolling and successive re-crystallization, which transformed the elastic residual strains into microplastic strains.

In regard to the peak exhibited by the cold-worked specimens in their 1st thermal run, it could be attributed to a dislocation-related relaxation, that is, to the bending of dislocations under the effect of the applied stress. Dislocation effects often appeared in plastically deformed alloys because of the high dislocation density. A number of pinning points (vacancy and atoms of different size in the Cantor’s alloy lattice) anchor dislocations, which bow under the applied stress. The result of dislocations bowing is, in many cases, a damping maximum. Specifically, it is tempting to attribute this low temperature peak to a Bordoni relaxation, since it exhibits some of the features typical of this relaxation [35]. A Bordoni relaxation can usually be fit as a Debye peak in the framework of the standard anelastic solid, as we were able to carry out for our experimental data. The peak appeared at a relatively low temperature. Finally, its strength was of the correct order. The detailed mechanism that gave rise to this peak involved a thermally activated formation of double kinks along the dislocations; however, despite many attempts, there is no complete agreement in the literature on the mechanism, therefore, in this case, further measurements would be necessary to remove uncertainty on the peak attribution. In any case, a treatment above 700 K was always enough to eliminate the peak, evidently since this temperature induced a microstructure relaxation that modified the dislocation pinning dynamics. This was evident in the data reported in Figure 2, where, apart from the peak, the sample damping capacity of the 2nd run was definitely below that of the 1st run.

The higher damping in the cold-worked samples, with respect to that of annealed sample, was due to the introduction of compressive residual stress during rolling. In addition, a higher dislocation density inside of the micro-twins, which was the plastic deformation mechanism of the Cantor HEA and the elongated grain boundaries (affected by cold rolling), contributed to this increase. Dislocation density and distribution control the overall damping of these alloys at low temperatures [36]. This was observed, as expected, in both the standard and the SLM specimens, as shown in Figure 2 and Figure 4. The twin formation inside of the FCC structure and their subsequent annealing were the main factors that changed the microstructure of the alloy.

### 4.2. High Temperature Peak of the CrNiCoFeMn Alloy Produced by SLM after Cold Rolling and Re-Crystallization

In a polycrystalline material, a possible source of attenuation is provided by the movement of grain boundaries. The effect was first proposed by Zener in 1941 [37], and relaxation phenomena occurring along grain boundaries have been studied since then by several authors [38]. The mechanism is that of an anelastic strain due to grain boundaries sliding between adjacent crystals under the action of shear stress. Triple points provide back stress, which restores the boundaries when stress is removed. This basic mechanism can be convoluted by the staking of dislocations at the grain boundaries, particularly in the case of cold-worked materials or when the sample thickness is of the same order of the grain diameter.

It has been suggested that precipitates can provide an alternative mechanism to triple points in grain boundary damping [39]. The proposed model is able to reasonably well describe the experimental data taken in several alloys containing precipitates and has a mechanical analogy with a dashpot with three elements in parallel, as follows: one spring for the total sliding strain due to elastic grain deformation, one dashpot for the intrinsic viscosity of a particle-free planar boundary and, finally, two dashpots in series for the sliding viscosity of grain boundaries containing particles, where the sliding across these particles is accommodated by boundary or volume diffusion [38]. As described more extensively in a previous work [22], fine σ phase (body-centered tetragonal lattice) precipitates were present in a specimen that was produced by SLM. These precipitates were clearly visible in the BSE high-resolution image of Figure 8b, together with even smaller precipitates that were visible in the SEM FEG high-resolution image of Figure 8c. The latter were nitrides, around which σ phase precipitates of a nanometric size grew. The presence of nitrides next to the sigma phase suggests that, at first, nitrides appeared, and then, by heterogeneous nucleation, the σ phase grew. These precipitates were absent, or in any case greatly reduced in number, in the standard samples. This was the main microstructural difference between the two kinds of sample.

The not-re-crystallized samples produced by SLM exhibited a damping peak just below 700 K. This peak could, therefore, be due to grain-boundary sliding controlled by the precipitates. The peak was, indeed, observed in the as-produced polycrystalline material and disappeared after the re-crystallization treatment, which brought about both a reduction in the grain boundary surface (grain growth) and in the number and dimension of the precipitates, as shown in Figure 8. Likewise, the peak was not observed in the standard Cantor’s alloy, where precipitates were rare or missing. The measured activation energy (127 kJ/mol) was compatible with that which is usually measured in grain boundary peaks. Nonetheless, the currently available data were not enough to unambiguously attribute the peak to a specific process, and further measurements will be carried out for this purpose in the future.

## 5. Conclusions


In this work, a mechanical spectroscopy study of two CrNiCoFeMn Cantor’s alloys obtained by induction melting (standard) and by selective laser melting (SLM), with the same lattice structure (FCC), was performed;Cold-worked samples of both kinds exhibited a damping peak at 400 K, with a relaxation strength of 2 × 10^−4^, and an apparent activation energy of 48 kJ/mol. The peak was tentatively attributed to a Bordoni relaxation, that is, to dislocation motion. It disappeared after a thermal treatment above 700 K;The SLM alloy samples alone exhibited a damping peak at 685 K, with a relaxation strength of 7.6 × 10^−4^, and an apparent activation energy of 127 kJ/mol. The peak was tentatively attributed to grain-boundary sliding controlled by precipitates (nitride and σ phase). The peak disappeared after a re-crystallization treatment;An exponentially growing damping background was measured in both the standard and the SLM samples, with typical values of about 3–5 × 10^−4^ at 300 K and 20 × 10^−4^ at 800 K;The room temperature (300 K) dynamic Young’s modulus of the SLM samples was rather low, (90 ± 10) GPa, in the not-re-crystallized case, due to porosity. It grew to (170 ± 10) GPa after re-crystallization treatment.


## Data Availability

Data available on request.

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
