# Peer review of "Mechanical Spectroscopy Study of CrNiCoFeMn High-Entropy Alloys"

_materials, 2023, doi:10.3390/ma16103689_

Round 1
Reviewer 1 Report
Several reservations are available in the manuscript that led it toward te major revision:
1-In the abstract, it is written that the HEA is developed through induction heating and SLM, While the last paragraph of the introduction two conditions "deformed 78 by cold rolling and re-crystallized" are written, might these are the post-treatment, but it's not synchronized with the abstract.
2- The introduction section needs to be oriented in the direction of the work performed by the authors.
3- What is meant by mechanical spectroscopy, explain it in the introduction to increase the readability, because the whole manuscript is surrounded around it.
4- Place the actual picture of specimens fabricated through SLM and Induction heating.
5- Include subheadings for the Materials and Methos section, like SLM specimens, Induction heating, post-treatment, mechanical testings, etc.
6- I think this manuscript is extracted from the thesis, that's why Chapter 4 and Chapter 1 are written at lines 174 and 179.
7- The same problem of mentioning the chapter numbers is available in several places.
8- Kindly include the legends for all the Figures, what black, red, and brown colors are representing?
9-Label the x and y axes of the graph and figures.
10- Arrange Figure 10 properly.
The manuscript is poorly configured in terms of writing and arranging the different sections. First, an extensive revision is required to shapeup the article for consideration.
Tuning of the sentences is required.
Author Response
Dear referee,
In the attached file You will find the answers to all your comments and suggestions.
Sincerely Yours,
Enrico Campari

Reviewer 2 Report
Mechanical Spectroscopy of high entropy alloys were studied by Enrico et al. Selective additive manufacturing (SLM) process, starting from mechanically alloyed powders were compared and mechanical parameters such as Young's modulus was determined and compared by induction melting and SLM.
Abstract : It is very vague ,e.g. "The high entropy alloy (HEA) of equiatomic composition CrNiFeCoMn and with FCC crystal structure was prepared by either induction melting and additive manufacturing with a selective laser melting (SLM) process, starting from mechanically alloyed powders". What authors want to express is not clear. Abstract does not clearly explain the purpose of the study. It is not clear from results how Young modulus was determined.
Introduction: First two paragraphs are very basic information. pls combine it and write relevant information . The second portion which covers literature review is too small to establish the research gap and novelty.
I suggest writing the introduction section again and putting emphasis on the aim of the study. There are long sentences with multiple commas which will divert the readers. I suggest making precise and short sentences.
It is recommended to add a schematic illustration as presently it is very overall very confusing.
Change figures to capital Figure .Correct throughout the manuscript.
Result : Why only one mechanical parameter is determined YM?
Line 172-173 exponent is missing.
Line 179: [19,chapter 1 ] is it reference?
Line 211:correct 2x10-4
what is the reason for 300K as reference temperature? Figure 6 is from one of the author’s previous works. What is the purpose of adding this figure ,and how it is relevant here please explain.?
Figure 8 is scattered. What is boundar (b)?
I would suggest explaining Figures again comparing it with the relevant references instead of putting them.
adjust Table 1 closer to it explanation.
In the abstract it was mentioned that Cr rich precipitates were seen. I didn't see anything relevant in the result section.
How was it established that dislocation and grain boundaries have certain impact during any cold work or recrystallization ? Please explain
Conclusion : It is not consistent with the abstract e.g It was mentioned that measurements were compared at 300K, but the conclusion is totally different.
Overall : I would suggest adding more relevant and useful results as current result and discussion section is quite superfluous. I would recommend major revision.
Author Response
Dear referee,
in the uploaded file You will find the answer to all your comments and suggestions.
Sincerely Yours,
Enrico Campari

Reviewer 3 Report
This manuscript focuses on the mechanical characterisation of a high entropy alloy (CrNiCoFeMn) using two fabrication approaches – conventional (induction) melting and selective laser melting. High entropy alloys are a hot topic, and these metallic materials display interesting combination properties, making them promising candidates to replace conventional alloys in some applications. The manuscript is well written but, in my opinion, there are a few questions that should be addressed. As an overall note, the authors present many results that are not original to this work, but retrieved from a previous article, treating them as if they were original which should not be done. I suggest authors to focus on the MS results’ analysis and mention in the introduction the importance of knowing the damping and elastic properties of the alloy relative to its prospective applications.
Technical/scientific:
1- Line 117, as-cast specimens were not subjected to mechanical spectroscopy.
2- Line 127, the EBSD dataset processing should be indicated. However, no original EBSD results were presented in the manuscript. The same can be said about SEM and XRD. The authors should not present as experimental work that is not original to this manuscript.
3- Line 169, chart colours have been changed. It is easier to read the charts if a legend is added and a tag identifying the manufacturing process and condition.
4- Line 174, how many trials were done for each specimen and how many specimens were used of the same condition?
5- Line 278, how do you know the precipitates were already present in the material?
6- Line 279, if the precipitates did not provide sufficient scattering volume, how did you detect/identify them?
7- Line 283, the precipitates should be identified in the image. Also, how can you identify and distinguish them between different conditions by optical microscopy? What about the microstructures from specimens produced by induction melting? These should be presented. A chemical (EDS would at least suffice) should be presented for each manufacturing process.
8- Figure 6, the legend is messed up.
9- Figure 8, these results (along with XRD) should not be presented in this manuscript. Even though authors state it is reprinted from a previous work, these are not original to this work.
10- Line 396, the damping peak near 400 K is relative to induction melting manufactured specimens only, not cold-worked samples in general. This should be clarified.
Formatting:
1- Line 62, there is no need to capitalise the mechanical properties in the middle of a sentence.
2- Line 313, for consistency, it is better to preserve the element sequence when identifying the alloy (CrNiCoFeMn).
Author Response

(The authors gave the same response as above.)

Round 2
Reviewer 1 Report
The authors have not entertained Question 4. It was requested to add the images of prepared specimens after SLM manufacturing, and induction hardening. The authors have referred to Fig. 7a and 7b which are the optical microscope images, which is unacceptable.
Authors also need to check throughout the manuscript for small mistakes and typo errors.
requires fine tuning
Author Response
Dear referees,
In the following, it is reported the list of your further comments and suggestions, together with our answer to them. The article has been modified accordingly.
Sincerely Yours,
Enrico G. Campari
Angelo Casagrande

Reviewer 2 Report
I would suggest adding better resolution of all Figures. They are blur. This coud be due to conversion in pdf. But please make sure figures of high quality. Otherwise, alll the comments have been addressed properly and paper can be accepted.
Author Response

(The authors gave the same response as above.)

Reviewer 3 Report
Most of my concerns have been addressed. However, as I previously highlighted, you state in Conclusions that “cold-worked samples exhibited a damping peak at 400 K”. However, based on Figures 2 and 5, cold-worked standard alloy presents a peak at 400K, whereas cold-worked SLM alloys display a peak near 700K. Additionally, I cannot see magenta in your charts. What you call magenta looks dark red/maroon to me.
Author Response

(The authors gave the same response as above.)
